# Effect of Loading Methods on the Fatigue Properties of Dissimilar Al/Steel Keyhole-Free FSSW Joints

**DOI:** 10.3390/ma13194247

**Published:** 2020-09-23

**Authors:** Zhongke Zhang, Yang Yu, Huaxia Zhao, Hui Tong

**Affiliations:** 1School of Materials Science and Engineering, Lanzhou University of Technology, Lanzhou 730050, China; 2School of Materials Science and Engineering, Tongji University, Shanghai 201804, China; 3Aeronautical Key Laboratory for Welding and Joining Technologies, AVIC Manufacturing Technology Institute AECC, Beijing 100024, China; zhaohuaxia@cfswt.com; 4SAIC MOTOR Commercial Vehicle Technical Center, Body Department, Shanghai 200090, China; tonghui01@saicmotor.com

**Keywords:** FSSW, dissimilar metals, fatigue properties, fracture, hysteresis loop

## Abstract

The energy evolution, fatigue life and failure behaviour of dissimilar Al/steel keyhole-free Friction stir spot welding (FSSW) joints were studied under different fatigue loads. The absorption energy of fatigue fracture, the fracture mechanism and the sensitivity of the fatigue limits to the fatigue load parameters were analysed. It was found that the stress ratio R determines the fatigue limit Ff, while the fatigue limit F_f_ is not sensitive to the loading frequency. The high-frequency fatigue load will increase the displacement deformation μ and fatigue fracture absorption energy E_a_ of the spot-welded joint, which are larger under asymmetric fatigue loading than those under symmetrical fatigue loading. At the same time, the symmetrical fatigue load can form the steady-state hysteresis loop, while asymmetric fatigue loading cannot, but asymmetric fatigue loading exhibits the displacement increment of fatigue softening. The fracture failure of spot-welded joints is a multiple crack source and the mixed-mode of ductile and brittle fracture mechanism, which exhibits typical fatigue striations in the fatigue fractures.

## 1. Introduction

With the proposed concept of energy conservation and emission reduction, lightweight technology is considered to be the most direct and effective means to achieve this goal. Al alloys, Mg alloys and other light metal materials have been widely considered to replace steel, especially in the automotive field. The problems of connecting dissimilar metals, such as Al/Mg [1], Al/Cu [2], Al/steel [3] and Mg/steel [4], are discussed. The main problem is that the melting points of dissimilar metals are different, and the welding metallurgy and metal weldability are also different. It is easy to produce welding defects in the joint, such as porosity, inclusion, brittle phases, and welding hot cracks, especially in fusion welding. This seriously limits the application of Al, Mg and other light metals in the structural integration of dissimilar metals [1,2,3,4].

Friction stir welding (FSW) is a solid phase connecting technology invented by the welding institute (TWI) in 1991. It is widely used in the welding of similar and dissimilar metals. Since there is no melting of metal during the welding process, the metal is completely in the thermoplastic state, so it is called the solid phase welding process. The friction between the shoulder of the welding tool and the workpiece can remove the stubborn oxide film on the surface of the metal and solve the problem of the oxide film of aluminium alloy forming inclusions in the weld. Compared with the traditional welding method, this technology has the advantages of high joint quality, small welding deformation, low residual stress, a green welding process and no pollution. It is widely used in the aerospace, shipping, and transportation industries and other fields [3,4,5,6].

Friction stir spot welding (FSSW), as a new welding method of FSW, is a new solid-state welding process combining FSW technology and spot welding joints as the main form. Traditional FSSW will leave a keyhole of the size of a pin in the joint, which undoubtedly reduces the strength of the joint and seriously affects the mechanical properties. To improve the mechanical properties of FSSW joints, Yoon et al. adopted swept FSSW AL to expand the connect area and transfer the keyhole position, which improved the strength of the joint to a certain extent [3]. Zhang [5] and Dong [6] et al. welded Al and steel by using retracted FSSW and refilled FSSW, respectively, which successfully eliminated the keyhole defects of the joints, which not only improved the appearance but also enhanced the mechanical properties of the joint. The keyhole-free FSSW is expected to replace riveting to achieve the same strength of connection performance, which can be widely used in the connection of the aircraft envelope. At the same time, FSSW is easy to realize automation, thus shortening the construction period. Similarly, the resistance welding technology of automobiles can also be replaced by FSSW technology. The Al/steel joint easily forms intermetallic compounds (IMCs) at the interface, which can be controlled by the heat input [7,8,9]. Bozzi et al. pointed out that there is an optimum thickness of the IMC layer [10]. The mechanical connection of the hook is a typical feature of FSSW joints [4,11]. The key factors affecting the formation of the hook are the presence or absence of the pin [12,13], the shape of the pin [14] and the insertion depth [15]. These welding parameters affect the microstructure [16], mechanical properties [17] and fracture mode [6] of joints. Sajed and Bisadi presented the concept of the effective distance (ED) to explain the reason for the different strengths of joints that are welded with different parameters and from different materials [18].

Zhou et al. found that the fatigue life of FSW in 5083 Al alloy is 9~12 times longer than that of MIG-pulse welds under a stress ratio of R = 0.1 [19]. Uematsu et al. also found that the fatigue strength was relatively higher in FSSW joints than in resistance spot welding (RSW) joints and that the tensile strength and fatigue strength of FSSW joints of Al alloy with or without keyholes showed the opposite result [20,21]. Therefore, the fatigue failure mechanism and detection and evaluation of FSSW joints are important research topics for their application. Malafaia et al. successfully monitored and evaluated the dynamic defects of FSSW joints under fatigue test conditions by the comparative monitoring vacuum (CVM) method, but only on joints with keyholes [22]. Joy-A-Ka and Ahmadi used Haibach’s method and a strain-based approach, respectively, to more accurately evaluate the fatigue damage and life of FSSW joints [23]. Grujicic et al. pointed out that fatigue data associated with FSW joints are characterized by a relatively large statistical scatter [24]. This scatter is closely related to the intrinsic variability of the FSW process and to the stochastic nature of the microstructure and properties of the workpiece material as well as to the surface condition of the weld. The structural integrity [25], IMCs and accumulation of dislocations [26] and nonuniform deformation [27] will affect the fatigue life and failure behaviour of FSW joints in different ways. Similarly, the failure modes are also different under different fatigue loads [1,28]. The failure mode of nugget pull-out is often found in FSSW joints [29]. The crack is always initiated at the crack tips near the unwelded joint interface [30] and at the defects of the subsurface in the weld nugget zone [31,32]. Kakiuchi et al. evaluated fatigue crack propagation (FCP) in dissimilar Al/steel FSW and found that the FCP rates were comparable or slightly faster than those of the Al base metal [33]. Zhang et al. found that microstructural inhomogeneity and crack closure easily contribute to the fluctuation FCP rate. A novel localized plasticity method, cold expansion, could improve the fatigue life of FSSW joints in all load ranges [34]. Hassanifard et al. found that it could improve the fatigue life up to six times in high cycle regimes [35].

At present, international research on the fatigue performance of FSSW joints is mainly focused on fatigue life, fatigue load, failure mode and damage assessment. There are few reports on the fatigue energy evolution process, especially for dissimilar Al/steel keyhole-free FSSW joints. In this work, the energy evolution process and failure behaviour of dissimilar Al/steel keyhole-free FSSW joints under different fatigue loads are studied to determine the absorption energy of fatigue fracture, and the sensitivity of the fatigue life to fatigue load parameters is analysed.

## 2. Experimental Procedures

### 2.1. Experimental Materials

In this experiment, AA6082-T6 aluminium alloy and a DP600 galvanized steel plate were used, and the sizes of the plates were 150 × 50 × 2 mm^3^ and 150 × 50 × 1 mm^3^, respectively. Their chemical compositions are listed in Table 1.

### 2.2. Welding Methods

Retracted pin technology is used in dissimilar Al/steel keyhole-free FSSW to eliminate the keyholes of joints. The welding tool adopts a pin-shoulder separation structure made of a nickel-based superalloy. The dissimilar Al/steel keyhole-free FSSW joints adopt the lap method of steel on aluminium.

Figure 1 shows the schematic diagram and flow chart of the dissimilar Al/steel keyhole-free FSSW process [5]. The welding parameters are listed in Table 2 [5]. In the process of dissimilar Al/steel keyhole-free FSSW in Figure 1a, the welding tool first rotates and plunges at speeds of ω and v_p_, respectively (as shown in Figure 1a). When the shoulder reaches the surface of the workpiece, the welding tool continues to plunge down, and the plunge depth of the shoulder is l_p_ (as shown in Figure 1b). After welding for a period of time, the welding tool starts to move forward at speed v, and at the same time, the pin is retracted back at speed v_b_ (as shown in Figure 1c). This is equivalent to the welding tool moving forward by a distance D, which makes the thermoplastic metal at the front of the welding backfill into the keyhole at the rear, so as to eliminate the keyhole. At the same time, the workpiece table slowly rotates in the opposite direction at a rotation speed ω_t_ until the pin is completely retracted to obtain a better keyless-free spot welding joint with a larger connection area. This is because as the pin moves forward, the rotation of the workpiece table causes the connection area to expand from a micro line segment to a large circle. It is worth noting that the three actions of the forward and retraction of the pin and the reverse rotation of the workpiece table should be carried out at the same time. Then, the workpiece table stops rotating, and the welding tool moves upward (as shown in Figure 1d). When the welding tool moves to the safe location, the shoulder and pin are simultaneously reset (as shown in Figure 1e). At this time, a dissimilar Al/steel keyhole-free FSSW joint is obtained (as shown in Figure 1f). The flow chart of the dissimilar Al/steel keyhole-free FSSW process is shown in Figure 1g. The whole welding process can be summarized as follows: the welding tool first rotates and plunges. When the shoulder touches the workpiece, the welding tool moves forward, the pin is retracted, and the worktable rotates reversely. When the welding tool became pin less state, the welding ends. The welding flow follows the welding process and the process control program is based on the flow chart.

### 2.3. Tensile-Shear Test

To determine the fatigue test parameter, according to the ASTM E8 standard, the tensile-shear test of dissimilar Al/steel keyhole-free FSSW joints was carried out by a WDW-100D electronic tensile testing machine (Five Star Testing Instrument Co., Ltd., Jinan, China) with a tensile speed of 0.2 mm/min. The corresponding thickness gasket was added at the clamping end of the sample to ensure that the spot welding sample was vertical in the tensile-shear test. The load–displacement curve was recorded and displayed in real-time on a computer terminal. Each group was tested three times to eliminate test errors. Figure 2 shows the drawing and load–displacement curve of the tensile-shear specimen of dissimilar Al/steel keyhole-free FSSW joints. The tensile-shear strength of the spot-welded joint with 1000 rpm is the highest at approximately 9 kN. This is because the tensile-shear strength of the spot welding joint is directly related to the interface morphology, defects, IMCs and other microstructure, which are affected by the welding process. The welding rotation speed directly affects the welding temperature. If the temperature is too low, it cannot be effectively connected. If the temperature is too high, too many IMCs will be formed. The joint at 1000 rpm has good plasticity, toughness and crack propagation resistance [5].

### 2.4. Fatigue Test

The fatigue test of dissimilar Al/steel keyhole-free FSSW joints was carried out by a 512A dynamic stiffness fatigue testing machine (Letry Material Testing Technology Co., Ltd., Xi’an, China). The specimen was fixed by the clamping method of the tensile specimen, keeping the joint specimen vertical and parallel to the loading force direction. The sine wave load and load control mode were adopted by referring to the ASTM E466 standard. The test data were recorded in real-time on a computer terminal, and the load–time, displacement–time and load–displacement curves were also displayed. Each group of tests was conducted three times to eliminate test errors. The fatigue fracture of spot welding joints was studied on a JSM-5600LV low-vacuum scanning electron microscope (SEM) (Japan Electron Optics Laboratory Company, Tokyo, Japan), and the element distribution was measured with an X-ray energy-dispersive spectrometer (EDS) (Oxford Instruments, Oxford, UK).

The fatigue test parameters of dissimilar Al/steel keyhole-free FSSW joints are set based on the mechanical properties of the tensile shear tests, and the joints with optimal parameters are selected as the fatigue research object. Because the tensile shear strength of dissimilar Al/steel keyhole-free FSSW joints with welding parameters of 1000 rpm is the highest, approximately 9 kN, the samples with welding parameters of 1000 rpm are selected as the test target of the fatigue test. To be conservative, the maximum tensile shear strength Fb for the fatigue test is set to 8 kN. The load form of the fatigue loading is selected to be a sine wave load. The frequencies are 20 Hz at a low frequency and 100 Hz at a high frequency. The stress ratios R are 0.1 and −1 to simulate the tension–tension and tension–compression fatigue tests, respectively. The low-frequency asymmetric fatigue test uses a sine wave load with a vibration frequency f = 20 Hz and a load ratio R = 0.1 to perform a tension–tension fatigue test on the spot welded joint. The high-frequency asymmetric fatigue test uses a load with a vibration frequency f = 100 Hz and a load ratio of R = 0.1 to perform a tension–tension fatigue test. The low-frequency symmetric fatigue test uses a load with a vibration frequency f = 20 Hz and a load ratio R = −1 to perform a tension–compression fatigue test. The fatigue test parameters of asymmetric fatigue loads (R = 0.1) and symmetric fatigue loads (R = −1) of dissimilar Al/steel keyhole-free FSSW joints are listed in Table 3.

## 3. Results and Discussion

### 3.1. Formation of the Joint

Figure 3 shows the macroscopic morphology and diagram of the formation of dissimilar Al/steel keyhole-free FSSW joints. Figure 3a shows that the keyhole is eliminated on the spot welded joint. The appearance of the spot welding joint is smooth, the surface of the welding spot is relatively flat, and the shape is has a good appearance. According to the welding process and fracture morphology, the spot welding joints can be divided into four areas: the stirring zone (STZ), shoulder affected zone (SAZ), non-affected zone (NAZ) and non-welded zone (NWZ), as shown in Figure 3b. With pin stirring, the thermoplastic metal in the STZ forms a vortex flow field [36], as shown by the black arrow in Figure 3b, which will form a cloud-like microstructure [37]. Under the pressure of the shoulder, the thermoplastic metal flows outward along its surrounding gap [38], forming an intermetallic compound transition layer at the interface [5], as shown by the coloured interface on the SAZ and NAZ in Figure 3b. The transition layer is mainly composed of FeAl_3_ phase, which is formed by thermal diffusion and the metallurgical combination of mechanical extrusion [5].

### 3.2. F-N Curve of Fatigue Loading

Figure 4 shows the F-N curve of fatigue loading (load-life curve of fatigue) of dissimilar Al/steel keyhole-free FSSW joints. Figure 4 shows that the fatigue life of the spot-welded joint increases with decreasing fatigue load. The fatigue life of the spot-welded joint tends to be infinite until the fatigue load reaches below the fatigue limit F_f_. At the same time, it can be clearly seen that when the loading frequency is constant, the fatigue limit F_f_(R = −1) (abbreviated as F_−1_) of the spot-welded joint under the symmetric load is less than the fatigue limit Ff under the asymmetric load. When the stress ratio R is constant, the fatigue limit F_fH_ under high-frequency loading is almost the same as the fatigue limit F_fL_ under low-frequency loading. The only difference is the speed required to reach the fatigue limit F_f_. The higher the frequency is, the faster the speed of approaching the fatigue limit. This is because symmetrical loading will increase the fatigue hardening degree of spot-welded joints and accelerate the fracture failure process of spot-welded joints, thus reducing the fatigue limit F_f_. However, the loading frequency only affects the fatigue fracture failure speed of spot-welded joints from the perspective of time and does not fundamentally affect the fatigue hardening and softening effect of spot-welded joints, so the loading frequency does not change the fatigue limit F_f_. In Figure 4, the fatigue limit F_−1_ of symmetrical loading is approximately 1.47 kN, while the fatigue limit Ff of asymmetric loading is approximately 2.24 kN. The difference between them is approximately 0.77 kN. It can be seen that the loading mode, that is, the stress ratio R, determines the fatigue limit F_f_, while the fatigue limit F_f_ is not sensitive to the loading frequency.

### 3.3. Research on the Energy Evolution of the Fatigue Damage Process

#### 3.3.1. Energy Evolution under Low-Frequency Asymmetric Fatigue Cyclic Loading

The actual displacement–time curve and load–displacement curve of dissimilar Al/steel keyhole-free FSSW joints with the maximum cyclic load F_max_ = 40%Fb are plotted for the low-frequency asymmetric fatigue test, as shown in Figure 5. Table 3 shows that the maximum cyclic load Fmax is 3.2 kN and the minimum cyclic load F_min_ is 0.32 kN. From the displacement–time curve in Figure 5a, it can be found that as the number of fatigue cycles increases, the spot welded joint also exhibits the same sinusoidal displacement fluctuation as the fatigue load. At the same time, the displacement value gradually increases until the spot welding joint fractures. The displacement fluctuation of the spot welding joint gradually develops from 0.1~0.02 mm to 0.22~0.09 mm, and its amplitude changes from 0.08 mm to 0.13 mm. The average displacement changes from the initial value of 0.06 mm to the final value of 0.155 mm, and the displacement increases by 0.095 mm, approximately 0.1 mm. This is the plastic deformation of the spot-welded joint during the fatigue loading process, as shown by the red line fitted in Figure 5a. The plastic deformation of dissimilar Al/steel keyhole-free FSSW joints is very small in the process of low-frequency asymmetric fatigue. With the increase in the number of fatigue cycles, the displacement amplitude increases relatively, and the displacement curve is trumpet-shaped, which is the manifestation of the stress relaxation of the spot welding joint caused by fatigue softening.

From the load–displacement curve in Figure 5b, it can be found that the spot welded joint does not form a steady-state hysteresis loop under low-frequency asymmetric fatigue loading but has a small amount of longitudinal displacement, and the displacement deformation is approximately 0.1 mm, which is the result of cyclic softening. Until the cycle is nearly 100,000 cycles, the spot welded joint fractures after reaching its fatigue life, and the fatigue test is stopped. Since the fatigue life is nearly 100,000 times, the load–displacement curve can be regarded as a slight continuous increase, almost covering the whole grey area in Figure 5b. The fatigue fracture absorption energy E_a_ of the spot-welded joint, that is, the fatigue crack nucleation and propagation work, can be determined by the geometric approximation method.

As shown in Figure 5b, the convex point of the load peak value is taken as the top edge CD of the parallelogram, and the concave point of the load valley value is taken as the bottom edge AB of the parallelogram. The left parallel line in the grey area is used as the left edge AD of the parallelogram, and the right BC of the parallelogram is equally divided into the right side of the grey area. The parallelogram ABCD is made. The bottom edge of the parallelogram has *μ_AB_* = 0.12 mm and the height is *F_h_* = 2.63 kN, so the fatigue fracture absorption energy *E_a_* of the spot-welded joint is
(1)$$E_{a}=F_{h}\times \mu_{AB}=2.63\times 0.12J\approx 0.32J\;$$

The energy required for low-frequency asymmetric fatigue fracture of spot-welded joints is very small, and a large part of the energy of the fatigue load is transformed into fatigue heat energy of the spot welding joints and dissipates. The fatigue heat energy is provided by the applied load, which is converted into the kinetic energy of the spot welding joint in the fatigue test. The kinetic energy is transformed into recoverable elastic deformation and nonrecoverable plastic deformation of the spot welding joint. Therefore, the volume heat, that is, the fatigue heat energy, is generated.

#### 3.3.2. Energy Evolution under High-Frequency Asymmetric Fatigue Cyclic Loading

The actual displacement–time curve and load–displacement curve of dissimilar Al/steel keyhole-free FSSW joints with the maximum cyclic load of F_max_ = 40%Fb are plotted in the high-frequency asymmetric fatigue test, as shown in Figure 6. Similarly, as the fatigue cycle continues, the upper and lower peak values of the fatigue load remain almost unchanged and fluctuate at approximately 3.2 kN and 0.32 kN, respectively. From the displacement–time curve in Figure 6a, it can be found that the displacement value of the spot-welded joint also gradually increases. The displacement fluctuation of the spot welding joint gradually develops from 0.017~0.152 mm to 0.279~0.501 mm, and its amplitude changes from 0.135 mm to 0.222 mm. The average displacement changes from the initial 0.0845 mm to the final 0.390 mm, and the displacement increases by approximately 0.3 mm. This is the plastic deformation of the spot-welded joint under high-frequency asymmetric fatigue loading, as shown by the red line fitted in Figure 6a. The plastic deformation of dissimilar Al/steel keyhole-free FSSW joints is large in the process of high-frequency asymmetric fatigue loading, which is three times that under low-frequency asymmetric fatigue loading. With the increase in the number of fatigue cycles, the displacement amplitude increases slightly, and fatigue softening occurs, which results in the stress relaxation of spot-welded joints.

From the load-displacement curve in Figure 6b, it can be found that the spot welded joint does not also form a steady-state hysteresis loop under high-frequency asymmetric fatigue loading, which is consistent with low-frequency asymmetric fatigue loading. Under the condition of constant load control, the fatigue load-displacement curve has a large longitudinal offset caused by fatigue softening, and the displacement deformation is approximately 0.3 mm. The fatigue fracture absorption energy *E_a_* of the spot-welded joint can be determined by the geometric approximation method above.

As shown in Figure 6b, the parallelogram ABCD has a bottom edge of *μ_AB_* = 0.3066 mm and a height of *F_h_* = 2.46 kN, so the fatigue fracture absorption energy *E_a_* of the spot-welded joint is
(2)$$E_{a}=F_{h}\times \mu_{AB}=2.46\times 0.3066J\approx 0.75J$$

It can be seen that the energy required for high-frequency asymmetric fatigue fracture of spot-welded joints is large. This is because the spot-welded joints deform rapidly in the process of high-frequency fatigue loading due to the fast loading frequency. Most of the kinetic energy cannot be converted into fatigue heat energy but are converted into the elastic–plastic deformation energy of the spot welding joint to improve the kinetic energy conversion, so the fatigue fracture absorption energy *E_a_* of the spot welding joint is larger.

#### 3.3.3. Energy Evolution under Low-Frequency Symmetrical Fatigue Cyclic Loading

The actual displacement–time curve and load–displacement curve of dissimilar Al/steel keyhole-free FSSW joints with the maximum cyclic load of F_max_ = 20% Fb are plotted in the low-frequency symmetric fatigue test, as shown in Figure 7. Table 3 shows that the maximum cyclic load F_max_ is 1.6 kN and the minimum cyclic load Fmin is −1.6 kN. From the displacement time curve in Figure 7a, it can be found that the displacement value of the spot-welded joint also has a gradual increasing trend. The displacement vibration of the spot-welded joint gradually develops from −0.067~0.061 mm to −0.057~0.078 mm, and its amplitude changes from 0.128 mm to 0.135 mm. The amplitude increases relatively. The average displacement changes from the initial value of 0 mm to the final value of 0.05 mm, and the displacement increases by approximately 0.05 mm. This is the plastic deformation of the spot-welded joint under low-frequency symmetrical fatigue loading, as shown by the red line fitted in Figure 7a. The plastic deformation of dissimilar Al/steel keyhole-free FSSW joints is smaller in the process of low-frequency symmetrical fatigue loading, which is half of the displacement of the low-frequency asymmetric tension–tension cycle loading. With the increase in the number of fatigue cycles, the displacement amplitude slightly increases, and fatigue softening also appears, resulting in the stress relaxation of the spot welding joint.

From the load–displacement curve in Figure 7b, it can be found that a steady-state hysteresis loop is formed under low-frequency symmetrical fatigue cyclic loading. The hysteresis loop has a slight longitudinal offset caused by fatigue softening. Figure 7c shows the process of hysteresis ring offset. The fatigue fracture absorption energy *E_a_* ≈ 0.1 J is calculated by integrating the hysteresis ring area. The integral area of the final hysteresis loop is slightly smaller than that of the initial hysteresis loop, and the difference is approximately 0.01 J, which is the result of cyclic softening. At this time, the fatigue fracture absorption energy *E_a_* of the spot welding joint is slightly smaller than that of the fatigue parameter R = 0.1 because the tension–compression fatigue load is more likely to cause crack propagation than the tension–tension fatigue load, which easily causes spot welding joint fracture.

#### 3.3.4. Comparative Study on the Fatigue Properties under Different Fatigue Loads

Comparing the displacement–time curve and load–displacement curve of spot-welded joints in different fatigue parameter tests, the displacement deformation μ and fatigue fracture absorption energy *E_a_* of spot-welded joints in different fatigue parameter tests are listed in Figure 8. Figure 8 shows that a high-frequency fatigue load will increase the displacement deformation μ and fatigue fracture absorption energy *E_a_* of the spot-welded joint. This is because the faster the fatigue load frequency is, the more serious the elastic–plastic deformation of the spot-welded joint is. The more obvious the fatigue softening effect is, the greater the displacement variation μ of the spot-welded joint is. At the same time, the kinetic energy cannot be transformed into fatigue heat energy in time. Most of the energy is converted into the elastic–plastic deformation work and crack nucleation propagation work of the spot welding joints, so the fatigue fracture absorbing energy *E_a_* is also large. The deformation μ and fatigue fracture absorption energy *E_a_* of spot-welded joints under asymmetric fatigue load are larger than those under symmetrical fatigue load. This is because the tension-compression load weakens the fatigue softening phenomenon and enhances the fatigue hardening phenomenon, which accelerates fatigue failure behaviour. Therefore, the deformation μ and fatigue fracture absorption work *E_a_* of the spot-welded joint are smaller than those of the asymmetric load. At the same time, the symmetrical fatigue load can form the steady-state hysteresis loop; however, the asymmetric fatigue load cannot form the steady-state hysteresis loop, but it can form the displacement increment of fatigue softening.

### 3.4. Analysis of the Fatigue Failure Mode and Fracture Mechanism

#### 3.4.1. Macro Fatigue Fracture

Figure 9 shows the macroscopic morphology and diagram of fatigue fracture of dissimilar Al/steel keyhole-free FSSW joints. Figure 9a shows the macroscopic fracture morphology of the spot-welded joint under asymmetric fatigue loading. From the fracture of Al plate and steel plate, it can be seen that the sample is fractured at the joint. The Al plate and steel plate have light deformation. The macroscopic fatigue fractures on Al plate and steel plate are shown on the left and right sides of the fracture. Under fatigue load, the torsion deformation appeared from the initial stage I, as shown in the torsion stage II in Figure 9c, resulting in the normal stress along the normal direction of the interface. Under the normal stress, the joint tears along the interface gap and the crack propagates to the interior of the joint, as shown in tearing stage III in Figure 9c. The fracture morphology is as uneven as the tension shear fracture is produced [8], and there are long cracks at the bending part. The inhomogeneity is characterized by brittle fracture, which is mainly caused by the brittle phase of the intermetallic compound [5]. Figure 9b shows the macroscopic fracture morphology of the spot-welded joint under a low stress symmetrical fatigue load. It can be seen that the sample is fractured on the aluminium plate at the edge of the joint, almost without bending deformation.

#### 3.4.2. Micro Fatigue Fracture

Figure 10 shows the micro fatigue fracture morphology of dissimilar Al/steel keyhole-free FSSW joints. Figure 10a shows the fatigue fracture of the spot-welded joint under asymmetric fatigue loading. The fracture is extremely irregular, and the tearing layers are uneven in this area, which is consistent with the impact of fracture morphology [5]. The fracture has obvious brittle characteristics on the STZ, and its initiation sources are mainly brittle IMCs. The initiation of cracks occurs at multiple locations, and crack propagation extends inward along the circumference. The obvious dimple band structure can be seen on the SAZ, as shown in the upper left corner of Figure 10a, which exhibits plastic fracture characteristics. At the same time, it should be noted that there are typical fatigue striations on the STZ of the fatigue fracture, as shown in the lower-left corner of Figure 10a, which are not found in the tensile and impact fracture surfaces [5]. These fatigue striations have equal intervals and the same depth. This is because, in the process of cyclic fatigue load loading, the fatigue crack will move forward at a certain distance every time the fatigue load is applied. After unloading, there will be a crack path. Such repeated loading and unloading will form such typical fatigue striations with equal intervals and the same depth.

Figure 10b shows the crack propagation morphology of the aluminium side of the joint without fracture under low-stress symmetric fatigue loading. Circumferential crack propagation can be seen outside the joint, accompanied by shallow dimples on the SAZ and NAZ. Due to the existence of microcracks at the edge of the joint, the crack first propagates from the edge of the spot welding joint. With slow fatigue hardening under the low-stress symmetric fatigue loads, the crack continues to expand along the matrix until fracture. This shows that the fatigue strength of the spot-welded joint is better than that of the Al matrix under symmetric fatigue loading.

Further EDS analysis of the STZ at the yellow box position in Figure 10a is shown in Figure 11. It can be found in Figure 11 that Al, Fe, Zn, Mg, and Si elements are unevenly distributed at the fracture of the spot welding joint, and some elements are aggregated in some specific locations. Therefore, different intermetallic compounds exist in different areas of the fracture, and their composition and content are obviously different. Under the same experimental conditions, Zhang et al. [5] found that the joint was mainly composed of IMC FeAl_3_, AlZn_x_ and FeAl_3_Zn_x_, in which the hard-brittle phase was dominant. These brittle phases become the weak areas of the spot-welded joint on the STZ, causing crack initiation and propagation from there. Therefore, the fatigue fracture of dissimilar Al/steel keyhole-free FSSW joints is the fracture mechanism of multiple crack sources and the tough-brittle mixed mode.

#### 3.4.3. Analysis of Fatigue Failure

During the failure process of dissimilar Al/steel keyhole-free FSSW joints, the formation and propagation of cracks start from the weakest position, as shown in Figure 3b. The gap is the best location for crack initiation [37]. Since the metal is not completely filled in this area, the interface bonding is the weakest and the stress concentration is the largest. The crack will then continue to propagate into the joint due to the weak interface connections and brittle phases [38], as shown by the white arrows in Figure 3b. When the crack tip reaches the STZ, there are a large number of brittle phases and structural defects inside, such as microcracks, voids and stress concentrations [39].

Figure 12 shows the microcrack morphology at the interface of dissimilar Al/steel keyhole-free FSSW joints. It can be found in Figure 12 that there are many microcracks at the interface of the spot welding joint, and the cracks tend to extend from the interface to the inside of the steel. Comparing Figure 12a,b, it can be found that the shape of the microcracks is different. The crack shape is that of slender broken lines in Figure 12a, while it is fragmentary in Figure 12b, filled with fine metal particles inside. The linear microcrack is caused by severe deformation under shoulder extrusion during the welding process. At the same time, there is a large amount of mechanical combination in the joints, which is also the location of the linear cracks [5]. However, the fragmentary cracks are metal particles produced by pin for the cutting of the workpiece, which has worse crack growth performance. Such microcracks are very small, often between a few microns and dozens of microns, which is a serious internal defect in the spot welding joint. Under an external load, multiple cracks will propagate simultaneously and eventually fail. Plastic ductile dimples and brittle cleavage planes can be found in the fracture, which indicates that the fracture failure of spot-welded joints is a multiple crack source and the mixed-mode of ductile and brittle fracture mechanism, as discussed above [39]. In addition, unlike the micromorphology of tensile shear and impact fractures, there are typical fatigue striations that are not found in the tensile and impact fractures on the STZ.

## 4. Conclusions

The keyhole-free FSSW joints of dissimilar AA6082-T6 aluminium alloy and a DP600 galvanized steel sheet were studied. The tensile-shear strength of the joints with 1000 rpm was the highest at approximately 9 kN. The fatigue performance of the joint with 1000 rpm, including the energy evolution, fatigue F-N curve and fatigue fracture, was further studied under different fatigue loads. The conclusions are summarized as follows:(1)When the loading frequency is constant, the fatigue limit F_−1_ ≈ 1.47 kN of the spot-welded joint under the symmetric load is less than the fatigue limit F_f_ ≈ 2.24 kN under the asymmetric load. When the stress ratio R is constant, the fatigue limit F_fH_ under high-frequency loading is almost the same as the fatigue limit F_fL_ under low-frequency loading, but the speed of reaching the fatigue limit F_f_ is different. The higher the frequency is, the faster the speed of approaching the fatigue limit. Therefore, the loading mode, that is, the stress ratio R, determines the fatigue limit F_f_, while the fatigue limit F_f_ is not sensitive to the loading frequency.(2)The symmetrical fatigue load can form the steady-state hysteresis loop, while asymmetric fatigue loading cannot, but asymmetric fatigue loading can form the displacement increment μ of fatigue softening. At the same time, the high-frequency fatigue load can increase the displacement deformation μ and fatigue fracture absorption energy *E_a_* of the spot-welded joint, which are larger under asymmetric fatigue loading than those under symmetrical fatigue loading.(3)Under the low-stress symmetric fatigue load, the spot welded joint fractures on the Al plate at the edge of the joint, while it fractures on the STZ of the joint under asymmetric fatigue loading. The fracture failure of spot-welded joints is a multiple crack source and the mixed-mode of ductile and brittle fracture mechanism, which has typical fatigue striations on fatigue fractures.

## Figures and Tables

**Figure 1 materials-13-04247-f001:**
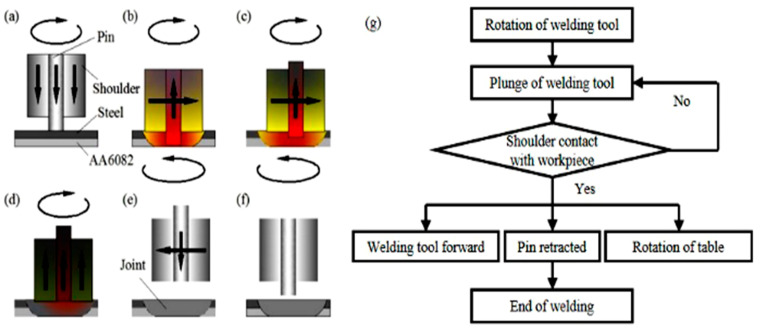
Schematic diagram and flow chart of dissimilar Al/steel keyhole-free FSSW process. FSSW), (**a**) welding initial stage, (**b**) warming-up stage, (**c**) welding stage, (**d**) welding end stage, (**e**) cooling stage, (f) initializing stage, (**g**) the flow chart of the keyhole-free FSSW process.

**Figure 2 materials-13-04247-f002:**
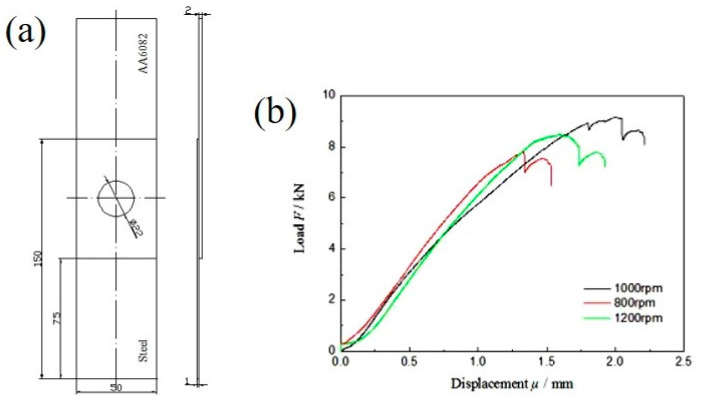
Drawing (**a**) and load-displacement curve (**b**) of tensile-shear specimen of dissimilar Al/steel keyhole-free FSSW joints.

**Figure 3 materials-13-04247-f003:**
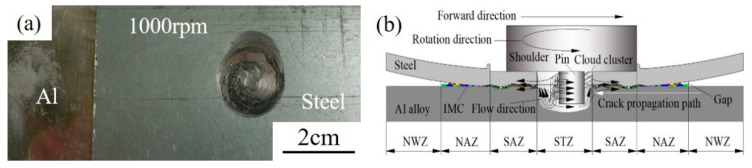
Macroscopic morphology (**a**) and diagram of formation (**b**) of Al/steel keyhole-free FSSW joints.

**Figure 4 materials-13-04247-f004:**
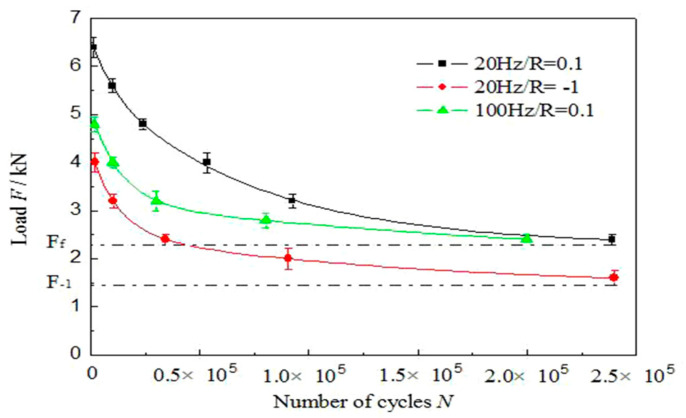
F-N curve of fatigue of dissimilar Al/steel keyhole-free FSSW joints.

**Figure 5 materials-13-04247-f005:**
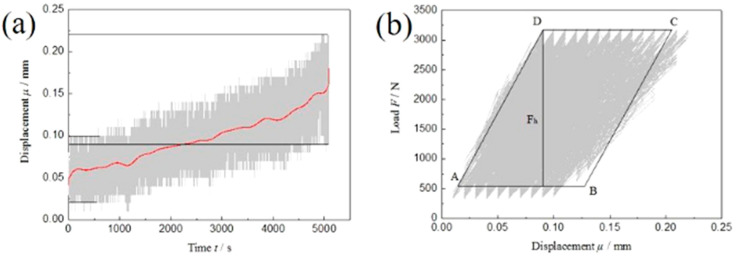
Displacement–time curve (**a**) and load–displacement curve (**b**) of low-frequency asymmetric cyclic load of dissimilar Al/steel keyhole-free FSSW joints.

**Figure 6 materials-13-04247-f006:**
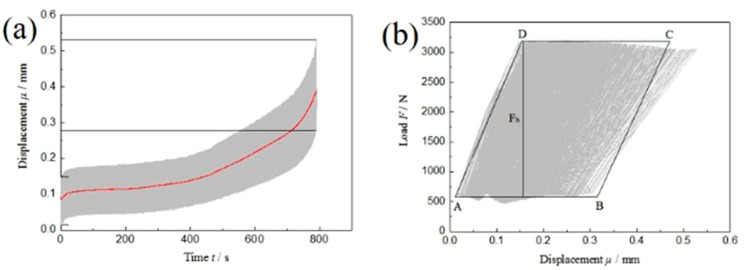
Displacement–time curve (**a**) and load–displacement curve (**b**) of high-frequency asymmetric cyclic load of dissimilar Al/steel keyhole-free FSSW joints.

**Figure 7 materials-13-04247-f007:**
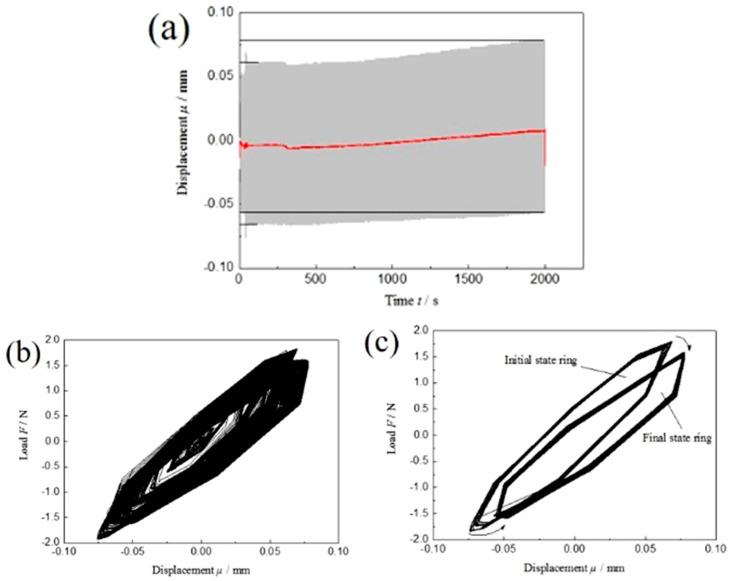
Displacement–time curve (**a**), load–displacement curve (**b**) and migration process of hysteresis loop (**c**) of low-frequency symmetrical fatigue cyclic load of dissimilar Al/steel keyhole-free FSSW joints.

**Figure 8 materials-13-04247-f008:**
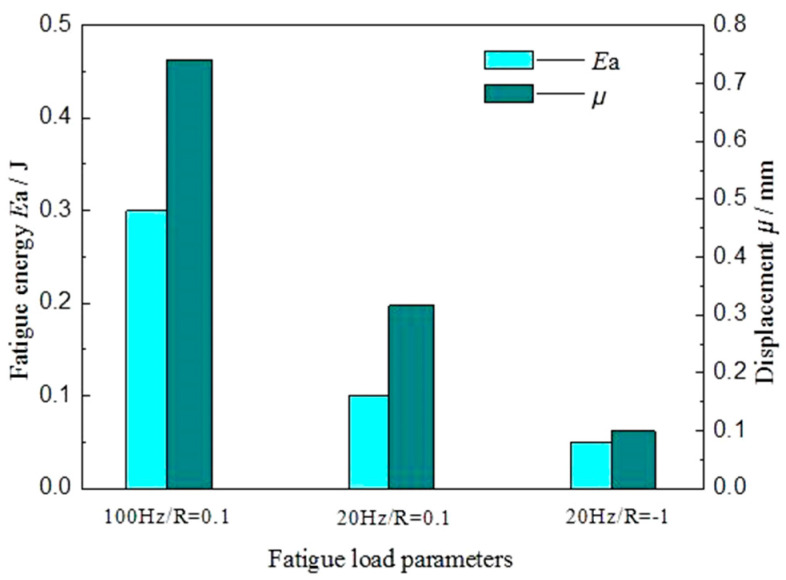
Histogram of displacement deformation and fatigue fracture absorbed energy of dissimilar Al/steel keyhole-free FSSW joints under different fatigue loads.

**Figure 9 materials-13-04247-f009:**
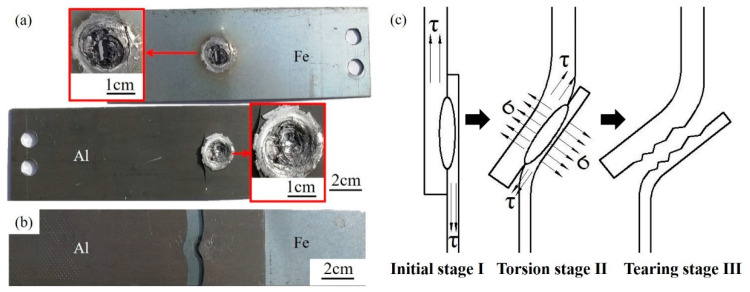
Macroscopic morphology (**a**,**b**) and diagram (**c**) of fatigue fracture of dissimilar Al/steel keyhole-free FSSW joints under asymmetric fatigue load (**a**) and under low-stress symmetric fatigue load (**b**).

**Figure 10 materials-13-04247-f010:**
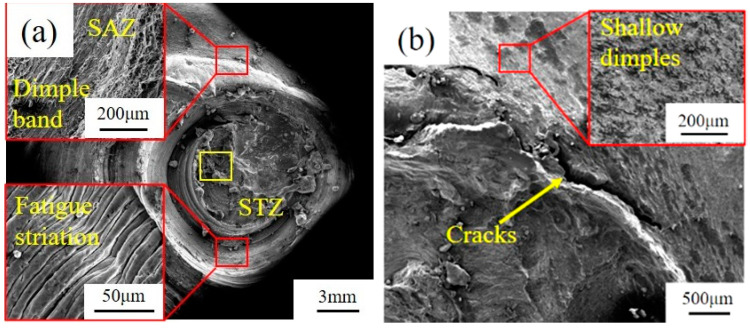
Micro fatigue fracture morphology of dissimilar Al/steel keyhole-free FSSW joints under asymmetric fatigue load (**a**) and under low-stress symmetric fatigue load (**b**).

**Figure 11 materials-13-04247-f011:**
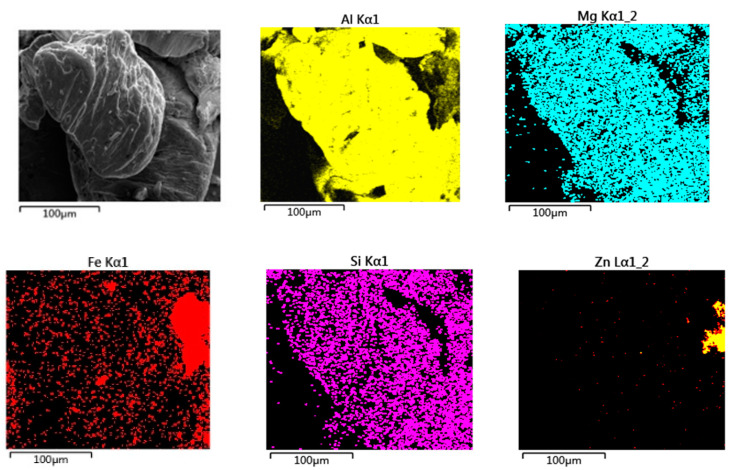
Energy-dispersive spectrometer (EDS) analysis of fatigue fracture on stirring zone (STZ) of dissimilar Al/steel keyhole-free FSSW joints.

**Figure 12 materials-13-04247-f012:**
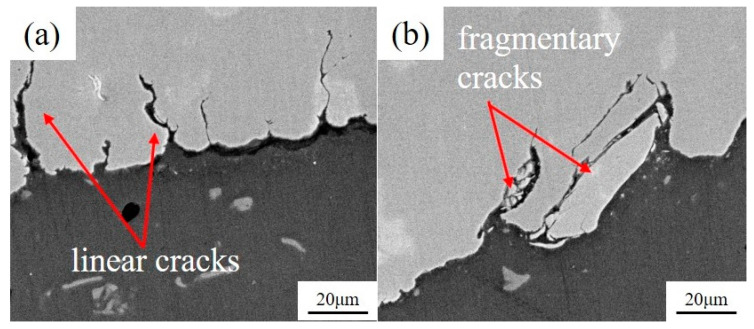
Linear cracks (**a**) and fragmentary cracks (**b**) at the interface of dissimilar Al/steel keyhole-free FSSW joints.

**Table 1 materials-13-04247-t001:** Chemical compositions of AA6082-T6 aluminum alloy and DP600 galvanized steel (wt%).

**AA6082**	**Si**	**Fe**	**Cu**	**Mn**	**Mg**	**Cr**	**Zn**	**Ti**	**Other**
Content	0.7–1.3	0.50	0.10	0.4–1.0	0.6–1.2	0.25	0.20	0.10	0.15
DP600	C	Mn	Si	Al	Mo	Cr	Cu	S	P
Content	0.09	1.84	0.36	0.05	0.01	0.02	0.03	0.005	0.005

**Table 2 materials-13-04247-t002:** Welding parameters of dissimilar Al/steel keyhole-free FSSW process.

**Pin Length***L* (mm)	**Plunging Depth***l*_p_ (mm)	**Shoulder Diameter***D*_s_ (mm)	**Retracting Speed***v*_b_ (mm/min)	**Rotation Rate of Welding** ω (rpm)
2.0	0.3	22.0	5.0	800/1000/1200
**Pin Diameter***D*_p_ (mm)	**Plunging Speed***v*_p_ (mm/min)	**Welding Speed***v* (mm/min)	**Forward Distance***D* (mm)	**Rotation Rate of Table** ω_t_ (rpm)
5.0	5.0	3.0	1.5	3.0

**Table 3 materials-13-04247-t003:** Fatigue test parameters of asymmetric fatigue load (R = 0.1) and symmetric fatigue load (R = −1) of dissimilar Al/steel keyhole-free FSSW joints.

**R = 0.1**	**80% F_b_**	**70% F_b_**	**60% F_b_**	**50% F_b_**	**40% F_b_**	**35% F_b_**	**30% F_b_**
F_maz_ (kN)	6.40	5.60	4.80	4.00	3.20	2.8	2.40
F_min_ (kN)	0.64	0.56	0.48	0.40	0.32	0.28	0.24
**R = −1**	**50% F_b_**	**40% F_b_**	**30% F_b_**	**25% F_b_**	**20% F_b_**		
F_maz_ (kN)	4.00	3.20	2.40	2.00	1.60		
F_min_ (kN)	−4.00	−3.20	−2.40	−2.00	−1.60

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
