# Peer review of "Effect of Loading Methods on the Fatigue Properties of Dissimilar Al/Steel Keyhole-Free FSSW Joints"

_materials, 2020, doi:10.3390/ma13194247_

Round 1
Reviewer 1 Report
The subject is worthy and interesting, and it is one to which the authors may add significant contributions, but the paper needs many changes and meticulously revised. To make this paper publishable the authors need to expand the introduction, rewrite few sections of the results, and add discussions. The literature review should be better presented discussing how the present contribution is a step forward to the existing research papers. The language of the manuscript should be refined; the paper would benefit from some closer proof reading as it includes some linguistic errors, e.g. toughness-brittle, etc.
I recommend the following revisions to be made before the paper can be considered for publication in Materials:
In the current manuscript (Introduction) the authors mentioned that FSW is " It is widely used in aerospace, shipping, and transportation industries and other fields [3-6]." This is a very general sentence the authors should also further explain why it is so important to investigate the keyhole-free FSSW topic; please emphasize its R&D relevance and industrial applications; bring examples please.
A general clear picture of the welding machine/apparatus should be added to Figure1; please elaborate on the welding process including the workpiece table rotation; in this context please explain the welding of relatively big products. The flow diagram shown in Figure 1 should be expanded and better explained.
In the Results and discussion chapter the authors stated: "174 …. forming an intermetallic compound transition layer at the interface [5], as shown by the coloured interface on the SAZ and NAZ in Fig. 3(b)." Please expand the explanations related to this subject tacking into account that no colored interfaces appear in Fig. 3(b).
Scale-bars are missing in Figure 1a to 1f, Fig. 3a and partially in Figs 9a,b.
Detailed information should be added and discussed regarding the accuracy and repeatability of the data shown in Figures 5 to 8; please add the pertinent standards used to prepare the specimens for both mechanical tests.
Figure 9 should be reorganized and please expand the explanations related to the macroscopic fatigue fracture morphologies at the low and higher magnifications.
Regarding "Figure 11. EDS analysis of fatigue fracture on STZ of dissimilar Al/steel keyhole-free FSSW joints." The authors stated that "367 .. Further EDS analysis on the STZ is shown in Fig. 11." The authors should quantify and elaborate on the metallurgical significance and on the repeatability of the EDS results; please show the location of measurements on pertinent metallographic pictures.
The authors mentioned that Figure 12 indicates "…Linear cracks (a) and fragmentary cracks (b) at the interface of dissimilar Al/steel keyhole-free FSSW joints." Please show the cracks' locations using pertinent metallographic pictures (cross section) and elaborate on the repeatability of the results.
My main reservations about the paper are related to the characterization of the studied welded samples. The authors are strongly recommended to analyze the welded parts by RT and or X-ray CT techniques and describe the size and location of the discontinuities (cracks, pores, etc.) found in the welded zones.
I hope above comments help to improve a future version of the paper.
Author Response
Dear Reviewer,
Thank you very much for your valuable advices.The comments were highly insightful and would help greatly to improve the quality of our manuscript. We have carefully reviewed and prepared the changes for each question. Attached are our response and revised manuscript.
Introduction
- In the current manuscript (Introduction) the authors mentioned that FSW is " It is widely used in aerospace, shipping, and transportation industries and other fields [3-6]." This is a very general sentence the authors should also further explain why it is so important to investigate the keyhole-free FSSW topic; please emphasize its R&D relevance and industrial applications; bring examples please.
Response: Thanks for the reviewer’s suggestions. The importance and industrial application of keyhole-free FSSW have been further explained in L50-L61. The relevant contents are as follows: “Traditional FSSW will leave a keyhole of the size of a pin in the joint, which undoubtedly reduces the strength of the joint and seriously affects the mechanical properties. To improve the mechanical properties of FSSW joints, Yoon et al. adopted swept FSSW AL to expand the connect area and transfer the keyhole position, which improved the strength of the joint to a certain extent [3]. Zhang [5] and Dong [6] et al. welded Al and steel by using retracted FSSW and refilled FSSW, respectively, which successfully eliminated the keyhole defects of the joints, which not only improved the appearance but also enhanced the mechanical properties of the joint. The keyhole-free FSSW is expected to replace riveting to achieve the same strength of connection performance, which can be widely used in the connection of aircraft envelope. At the same time, FSSW is easy to realize automation, thus shortening the construction period. Similarly, the resistance welding technology of automobile can also be replaced by FSSW technology.”
Experimental procedures
- A general clear picture of the welding machine/apparatus should be added to Figure1; please elaborate on the welding process including the workpiece table rotation; in this context please explain the welding of relatively big products. The flow diagram shown in Figure 1 should be expanded and better explained.
Response: Thanks for the reviewer’s suggestions. The welding process and flow diagram has been further explained in L112-L129 and L129-L131. The welding joint have a larger connection area, which is because as the pin moves forward, the rotation of the workpiece table causes the connection area to expand from a micro line segment to a large circle. It is worth noting that these three steps are carried out at the same time. It should be noted that the three actions of the forward and retraction of the pin and the reverse rotation of the workpiece table should be carried out simultaneously in order to realize the better keyless-free spot welding joint with a larger connection area.
The welding machine and welding tool are shown in the figure below, which is designed by ourselves. The schematic diagram in Fig. 1 can explain the process principle of the whole welding process, so the physical picture of the welding machine is not added. We worry that this will make the text lengthy.
Results and discussion
- In the Results and discussion chapter the authors stated: "174 …. forming an intermetallic compound transition layer at the interface [5], as shown by the coloured interface on the SAZ and NAZ in Fig. 3(b)." Please expand the explanations related to this subject tacking into account that no colored interfaces appear in Fig. 3(b).
Response: Thanks for the reviewer’s suggestions. The intermetallic compound transition layer is mainly composed of FeAl3 phase at the interface, which is formed by thermal diffusion and the metallurgical combination of mechanical extrusion, as shown by the coloured interface on the SAZ and NAZ in Fig. 3(b).
- Scale-bars are missing in Figure 1a to 1f, Fig. 3a and partially in Figs 9a,b.
Response: Thanks for the reviewer’s suggestions. Scale-bar has been added in Fig. 3a.
Figure 1a to 1f are schematic diagram, so scale-bars are not added. Fig 9a and 9b have the scale-bars in the lower right corner.
- Detailed information should be added and discussed regarding the accuracy and repeatability of the data shown in Figures 5 to 8; please add the pertinent standards used to prepare the specimens for both mechanical tests.
Response: Thanks for the reviewer’s suggestions. The fatigue test of dissimilar Al/steel keyhole-free FSSW joints was carried out by referred to the ASTM E466 standard. The test data were recorded in real time on a computer terminal, and the load-time, displacement-time and load-displacement curves were also displayed. Each group of tests was conducted three times to eliminate test errors. If there is a group of data with some big discrepancy, a group of experiments will be supplemented to improve the accuracy and repeatability of the data. This has been discussed in L154-L157.
- Figure 9 should be reorganized and please expand the explanations related to the macroscopic fatigue fracture morphologies at the low and higher magnifications.
Response: Thanks for the reviewer’s suggestions. the macroscopic fatigue fracture morphologies at the low and higher magnifications have been further explained in L350-L355 as follows: “From the fracture of Al plate and steel plate, it can be seen that the sample is fractured at the joint. The Al plate and steel plate have light deformation. The macroscopic fatigue fracture on Al plate and steel plate are shown on the left and right sides of the fracture. The fracture morphology is as uneven as the tension shear fracture is produced [8], and there are long cracks at the bending part. The inhomogeneity is characterized by brittle fracture, which is mainly caused by the brittle phase of intermetallic compound [5].”
- Regarding "Figure 11. EDS analysis of fatigue fracture on STZ of dissimilar Al/steel keyhole-free FSSW joints." The authors stated that "367 .. Further EDS analysis on the STZ is shown in Fig. 11." The authors should quantify and elaborate on the metallurgical significance and on the repeatability of the EDS results; please show the location of measurements on pertinent metallographic pictures.
Response: Thanks for the reviewer’s suggestions. The location of EDS measurements on STZ is shown at yellow box position in Fig. 10(a). the result of EDS analysis have further explained as follow: Al, Fe, Zn, Mg, and Si elements are unevenly distributed at the fracture of the spot welding joint, and some elements aggregated in some specific locations. Therefore, different intermetallic compounds exist in different areas of the fracture, and their composition and content are obviously different. Under the same experimental conditions, Zhang et al. [5] found that the joint was mainly composed of IMC FeAl3, AlZnx and FeAl3Znx, in which the hard-brittle phase was dominant.
- The authors mentioned that Figure 12 indicates "…Linear cracks (a) and fragmentary cracks (b) at the interface of dissimilar Al/steel keyhole-free FSSW joints." Please show the cracks' locations using pertinent metallographic pictures (cross section) and elaborate on the repeatability of the results.
Response: Thanks for the reviewer’s suggestions. Cracks often occur at the joint interface as shown in the figure below, which is caused by the uneven deformation under the large extrusion deformation of the mixing needle and the shaft shoulder. At the same time, there is a large amount of mechanical combination, which is also the location of the crack. This has been further explained in L411-L415. This crack is the data obtained in the experiment. The purpose of this quotation is to explain the reason of crack propagation. Therefore, the specific interface microstructure is not discussed, but its existence and repeatability are obvious.
- My main reservations about the paper are related to the characterization of the studied welded samples. The authors are strongly recommended to analyze the welded parts by RT and or X-ray CT techniques and describe the size and location of the discontinuities (cracks, pores, etc.) found in the welded zones.
Response: Thanks for the reviewer’s suggestions. This paper mainly studies the fatigue characteristics of spot welded joints. The microstructure of spot welded joints has been studied before. There is no systematic discussion here, but only cited as an explanation. It is a good suggestion from the reviewer to use RT and or X-ray CT technology to study the defects of joints, which can be considered as a characterization method for the next microstructure study. Thank you!
Some of the grammar, format, figures, explanation and discussion in this paper have also been modified and marked accordingly.
We would appreciate your efforts in reviewing our manuscript and shall look forward to hearing from your decision when it is made.
Sincerely
Yu
Tongji University
Reviewer 2 Report
This is an interesting work. However few clarifications are needed before publication:
- Fig. 2: Please provide adequate discussions as to why the shear strength is the highest at 1000 rpm?
- Line 170: Are these standard FSW terms? How about Heat affected Zone, Thermo-mechanically affected zone?
- Differentiate STZ and SAZ in Figure 3b.
- Line 186 187, use proper subscripts for Ffh and Ffl.
- Explain the noise reduction techniques and the errors associated with the techniques for analyzing the fatigue curves.
- What are the red pixels in the Zn map of Figure 11?
- Use the same notations for displacement in lines 266 and equation 2.
- Figure 12: Are these cracks closer to the "gap" in the joint? Where were the micrographs taken from relative to the sample?
- Figure 12 a and Figure 12b, the crack morphologies doesn't seem to be different. If the authors think that the morphologies are different, what are the implications of such crack morphologies?
Author Response
Dear Reviewer,
Thank you very much for your valuable advices.The comments were highly insightful and would help greatly to improve the quality of our manuscript. We have carefully reviewed and prepared the changes for each question. Attached are our response and revised manuscript.
- Fig. 2: Please provide adequate discussions as to why the shear strength is the highest at 1000 rpm?
Response: Thanks for the reviewer’s suggestions. The shear strength of spot welding joint is directly related to the interface morphology, defects, IMCs and other microstructure, which are affected by the welding process. The welding rotation speed directly affects the welding temperature. If the temperature is too low, it cannot be effectively connected. If the temperature is too high, too much IMCs will be formed. The joint at 1000rpm has good plasticity, toughness and crack propagation resistance. This has been further explained in L142-L147.
- Line 170: Are these standard FSW terms? How about Heat affected Zone, Thermo-mechanically affected zone?
Response: Thanks for the reviewer’s suggestions. The Heat affected zone (HAZ) and Thermo-mechanically affected zone (TMAZ), etc. are classified according to the microstructure. Here, the four areas, including the stirring zone (STZ), shoulder affected zone (SAZ), non-affected zone (NAZ) and non-welded zone (NWZ), are divided according to the welding process, as shown in Fig. 3(b). The purpose is to facilitate the discussion of the influence of the welding process on the joint.
- Differentiate STZ and SAZ in Figure 3b.
Response: Thanks for the reviewer’s suggestions. Fig. 3b has been modified, STZ and SAZ are shown in Fig. 3b.
- Line 186 187, use proper subscripts for Ffh and Ffl.
Response: Thanks for the reviewer’s suggestions. Ffh and Ffl have been modified to Ffh and Ffl. All notations have been checked and changed to the correct format.
- Explain the noise reduction techniques and the errors associated with the techniques for analyzing the fatigue curves.
Response: Thanks for the reviewer’s suggestions. The equipment uses the wavelet transform method processed by DSP to reduce the noise of the fatigue signal. The fatigue test of dissimilar Al/steel keyhole-free FSSW joints was carried out by referred to the ASTM E466 standard. The test data were recorded in real time on a computer terminal, and the load-time, displacement-time and load-displacement curves were also displayed. Each group of tests was conducted many times to eliminate test errors. If there is a group of data with some big discrepancy, a group of experiments will be supplemented to improve the accuracy and repeatability of the data.
- What are the red pixels in the Zn map of Figure 11?
Response: Thanks for the reviewer’s suggestions. Zn map only shows the distribution of Zn, no other elements.
- Use the same notations for displacement in lines 266 and equation 2.
Response: Thanks for the reviewer’s suggestions. The notations of displacement have been changed to μ in equation 1 and 2.
- Figure 12: Are these cracks closer to the "gap" in the joint? Where were the micrographs taken from relative to the sample?
Response: Thanks for the reviewer’s suggestions. Cracks often occur at the joint interface as shown in the figure below, which is caused by the uneven deformation under the large extrusion deformation of the mixing needle and the shaft shoulder. At the same time, there is a large amount of mechanical combination, which is also the location of the crack. This has been further explained in L411-L415. This crack is the data obtained in the experiment. The purpose of this quotation is to explain the reason of crack propagation. Therefore, the specific interface microstructure is not discussed, but its existence and repeatability are obvious.
- Figure 12 a and Figure 12b, the crack morphologies doesn't seem to be different. If the authors think that the morphologies are different, what are the implications of such crack morphologies?
Response: Thanks for the reviewer’s suggestions. The crack shape is that of slender broken lines in Fig. 12(a), while it is fragmentary in Fig. 12(b), filled with fine metal particles inside. The linear microcrack is caused by severe deformation under shoulder extrusion during the welding process. At the same time, there is a large amount of mechanical combination in the joints, which is also the location of the linear cracks [5]. However, the fragmentary cracks are metal particles produced by pin for the cutting of the workpiece, which has worse crack growth performance. It has been further explained in L411-L417.
Figure 12. Linear cracks (a) and fragmentary cracks (b) at the interface of dissimilar Al/steel keyhole-free FSSW joints.
Some of the grammar, format, figures, explanation and discussion in this paper have also been modified and marked accordingly.
We would appreciate your efforts in reviewing our manuscript and shall look forward to hearing from your decision when it is made.
Sincerely
Yu
Tongji University
Round 2
Reviewer 1 Report
I found that the authors have partially answered to my comments/questions and that the paper has been greatly improved. The paper would benefit from further closer proof reading as it still includes some linguistic errors.
I recommend some final revisions to be made before the paper can be considered for publication in Materials:
A general clear picture of the welding machine/apparatus should be added to Figure1; please elaborate on the welding process including the workpiece table rotation; in this context please explain the welding procedure when joining relatively big parts. The flow diagram shown in Figure 1 should be expanded and improved.
In the Results and discussion chapter the authors stated: "190 …. forming an intermetallic compound transition layer at the interface [5], as shown by the coloured interface on the SAZ and NAZ in Fig. 3(b)." Please further expand the clarifications related to this subject tacking into account that no colored interfaces appear in Fig. 3(b).
Scale-bars are missing in the inserts of Figs 9a,b.
Detailed information should be added regarding the pertinent standard used to prepare the specimens for the tensile-shear test; please add technical drawing of the specimens.
Please expand the explanations of the macroscopic fatigue fracture morphologies at the low magnifications (Figure 9) in terms of cracks macroscopic propagation and elaborate on the data obtained from the higher magnification results.
I hope above comments help to improve a future version of the paper.
Author Response
Dear Reviewer,
Thank you for the Editors’ comments for our manuscript “Effect of loading methods on the fatigue properties of dissimilar Al/steel keyhole-free FSSW joints” materials-932081 which was submitted to Materials. The comments were highly insightful and would help greatly to improve the quality of our manuscript. We have carefully reviewed and prepared the changes for each question. Attached are our response and revised manuscript.
We would appreciate your efforts in reviewing our manuscript and shall look forward to hearing from your decision when it is made.
Sincerely
Y. Yu
Reviewer 2 Report
The authors have addressed the comments adequately and the manuscript can be accepted for publication.
Author Response
Dear Reviewer,
Thank you very much for your review and comments. Some of the grammar, format, figures, explanation and discussion in this paper have been further modified and marked accordingly.
Sincerely
Y. Yu